# Exploring the Influence of IL-8, IL-10, Patient-Reported Pain, and Physical Activity on Endometriosis Severity

**DOI:** 10.3390/diagnostics14161822

**Published:** 2024-08-21

**Authors:** Ionel Daniel Nati, Andrei Malutan, Razvan Ciortea, Mihaela Oancea, Carmen Bucuri, Maria Roman, Cristina Ormindean, Alexandra Gabriela Milon, Dan Mihu

**Affiliations:** 12nd Department of Obstetric and Ginecology, “Iuliu Hatieganu” University of Medicine and Pharmacy, 400610 Cluj-Napoca, Romania; ionel.nati@umfcluj.ro (I.D.N.);; 2Emergency Military Clinical Hospital “Dr Constantin Papilian”, 400610 Cluj-Napoca, Romania; 3Faculty of Physical Education and Sport, Bogdan-Vodă University of Cluj Napoca, 400394 Cluj-Napoca, Romania

**Keywords:** endometriosis, severity, pain, interleukin

## Abstract

Endometriosis is known to be a chronic, debilitating disease. The pathophysiological mechanisms of endometriosis development include local chronic inflammation and a certain degree of local immune deficit. We investigated the relationship between the endometriosis severity, IL-8, IL-10, BDNF, VEGF-A serum and tissue levels, patient-related pain, and physical activity in a cohort of 46 patients diagnosed with endometriosis who underwent surgery. The same panel of biomarkers was investigated in a control group of 44 reproductive-aged patients with non-endometriotic gynecological pathology who underwent surgical intervention. Our data show a high statistical significance between tissue expression of IL-8, IL-10, patient-related pain, and the severity of endometriosis. No relationship was identified between serum or tissue levels of VEGF-A and BDNF and the severity of endometriosis. These results validate the presence of local chronic inflammation and immune deficit, thereby creating, alongside other studies in the field, an opportunity for the development of innovative and personalized treatment approaches in endometriosis.

## 1. Introduction

Endometriosis is defined as the presence of active endometrial glands and stroma outside the uterine cavity. It is a benign, chronic, and debilitating condition, and the progression depends on a series of risk factors and individual factors. The true incidence of endometriosis is not yet known due to the polymorphism of the clinical picture, which makes diagnosis difficult. There are situations where symptomatic patients present with forms of microscopic lesions that are unidentifiable by imaging, and conversely, asymptomatic patients despite having macroscopic lesions. According to the latest publications from the World Health Organization (WHO), endometriosis affects approximately 10% of women of reproductive age, making it one of the main causes of infertility [1,2,3,4].

Over time, a series of hypotheses have sought to explain the pathophysiology of endometriotic lesion development, but none have succeeded in fully explaining the process. Among the most appealing theories is that of retrograde menstruation, which explains the predominance of endometriotic lesions in the pelvic region but cannot account for unusual endometriotic lesions such as those in the lungs or brain [5,6,7,8,9,10]. The existence of a local immune deficit complements the first theory [11,12,13,14,15]. Other authors attribute the appearance of endometriotic lesions to lymphatic or vascular dissemination of endometrial tissue or celomic metaplasia [16,17,18,19].

The most common symptom of endometriosis that leads patients to seek medical attention is pain. Pain can manifest in various forms: menstrual pain, premenstrual pain, dyspareunia, and non-menstrual abdominal pain, with its localization also varying depending on the location of endometriotic lesions [20,21,22,23]. A very good quantification of pain is found within the questionnaire proposed by the British Society for Gynaecological Endoscopy (BSGE) [24]. The main questions included in the BSGE questionnaire are related to general questions about your pain (premenstrual pain, menstrual pain, noncyclical pelvic pain, Pain during sexual intercourse, etc.), Information about Bowel function, Medical Therapy, Fertility, painkillers, previous surgery for endometriosis and about your health in general. Additionally, it assesses the impact of pelvic pain on daily activities, emotional well-being, and overall quality of life. Also, endometriosis is among the most significant causes of infertility, being implicated in 30–50% of cases of female infertility [25,26,27].

The severity of endometriosis can be classified using systems such as the revised American Society for Reproductive Medicine (rASRM) staging system or Enzian classification [28,29,30,31]. In our study, a revised ASRM classification was applied. This system takes into account factors such as lesion size, extent, and involvement of pelvic structures to classify endometriosis into stages ranging from minimal (Stage I) to severe (Stage IV).

Diagnosis of endometriosis is sometimes challenging. Transvaginal ultrasound followed by magnetic resonance imaging (MRI) to map the lesions and extent of endometriosis most commonly enables accurate diagnosis. Laparoscopy allows direct visualization of lesions, biopsy for histopathological confirmation, and safe surgical treatment when necessary [32,33,34]. The importance of biomarkers has gained special interest in recent years. They can play a crucial role in early diagnosis, establishing the prognosis of endometriosis, monitoring treatment, identifying new therapeutic targets, and personalizing treatment [35,36,37].

Treatment of endometriosis can vary depending on the severity of symptoms, the patient’s age, desire for pregnancy, and other individual factors. It includes either medical management or surgical excision [38,39,40,41,42].

The prognosis of endometriosis depends on several factors, such as the severity of the disease, response to treatment, patient’s age, associated complications, and desire for conception. Its assessment is not strictly limited to the risk of lesion recurrence or the number of visits to the doctor required but encompasses a more detailed analysis of the quality of life of individuals diagnosed with endometriosis. It can be partially assessed through questionnaires proposed by BSGE. Current efforts are aimed at assessing prognosis based on serum or tissue levels of certain biomarkers [27,30,35,37,42,43].

### 1.1. Endometriosis Development and Biomarkers

In most circumstances, despite menstrual reflux into the peritoneal cavity, macrophages play an important role in eliminating these cellular debris, thus providing local protection. Macrophage concentration in peritoneal fluid seems to fluctuate during the menstrual cycle and is highest during menstrual bleeding. Estrogen (E2) has been found to be a regulator of type 2 macrophage phenotype (M2), which is responsible for its efficiency. These will secrete matrix metalloproteinase-9 (MMP-9) to break down the extracellular matrix, thus breaking down refluxed tissues into small pieces. Additionally, the CD-36 receptor (cluster of differentiation 36) is expressed on the cell membrane of macrophages to facilitate phagocytosis of these small fragments of endometrial debris. In the case of endometriotic lesions, increased concentration of prostaglandin E2 (PGE2) will suppress the expression of MMP-9 and CD36. This significantly inhibits the phagocytic function of macrophages, favoring the development of endometriotic lesions [44,45,46,47].

At the same time, macrophages, activated T lymphocytes, and even the endometriotic lesions themselves have the capacity to release a series of growth factors and pro-inflammatory and angiogenic cytokines that contribute to the development of endometriotic lesions:IL-8 (interleukin 8), known as an α-chemokine with chemotactic activity and acting as a strong angiogenic agent, primarily produced by peripheral macrophages, has been detected in high concentrations in peritoneal fluid of women with endometriosis. In addition to its chemotactic and activating properties for granulocytes, IL-8 has recently been found to stimulate the proliferation of endometrial cells [44,45,46,47].Vascular endothelial growth factor (VEGF), one of the main angiogenic factors with the ability to stimulate mitogenesis, migration, and differentiation of endothelial cells, is strongly expressed in endometriotic tissue as well as in peritoneal macrophages. Peritoneal-activated macrophages are a major source of VEGF in endometriosis, and this expression is directly regulated by ovarian steroids. E2 acts on various macrophage signaling pathways, especially those related to the support of inflammatory cell recruitment and the remodeling of inflamed tissues, such as mitogen-activated protein kinase (MAPK), phosphatidylinositol-3-kinase/protein kinase B, and Nuclear factor kappa-light-chain-enhancer of activated B cells (NF-kB). Consequently, dysregulation of ovarian steroid hormone homeostasis could influence the survival of ectopic endometrial cells and promote lesion vascularization. It is well known that hypoxia induces the expression of VEGF. Effects of hypoxia are mainly mediated by a protein complex called hypoxia-inducible factor-1 (HIF-1), which consists of two subunits, HIF-lα, the inducible unit, and HIF-1β, the constitutive unit. In endometriotic lesions, abnormally high levels of HIF-lα complex have been found, clearly proving that hypoxia is involved in the production of VEGF in endometriosis [46,47].IL-10 (interleukin 10), secreted largely by macrophages, is known for inhibiting T cell activation but also for reducing expression of co-stimulatory molecules (CD-80 and CD-86) and indoleamine 2,3-dioxygenase. Interestingly, macrophages secrete IL-15 (interleukin 15), a chemoattractant for uNK (uterine natural killer) cells, and downregulate the cytotoxicity of uNK cells [45,46].Nerve growth factors (NGF), especially BDNF (Brain-derived neurotrophic factor), are abnormally synthesized and released by activated macrophages, mast cells, NK (natural killer) cells, and leukocytes within endometriotic formations close to sensitive nerve fibers, and in peritoneal fluid. These sensitize or stimulate endings of sensitive nerve fibers, leading to a vicious cycle characterized by nociceptor sensitization, focal neoneurogenesis, and activation of sensitive nerve fibers, resulting in hyperalgesia [47].

### 1.2. Objectives

The primary objective of this study was to investigate the role of IL-8, IL-10, VEGF-A, and BDNF in assessing and determining endometriosis severity. On the other hand, we aimed to assess the relationship between the severity of endometriosis and the average pain score obtained through the application of the BSGE questionnaire to patients. Another objective is to evaluate the severity of endometriosis in relation to physical activity performed by the patient.

Measuring IL-8 levels can help identify the inflammatory and angiogenic activity associated with endometriosis, aiding in early detection. IL-10’s role in immune modulation highlights its potential as a marker for immune response in endometriosis, guiding therapeutic strategies. BDNF’s involvement in nerve sensitization and pain underscores its importance in diagnosing pain-related symptoms of endometriosis and developing targeted pain management treatments. VEGF is crucial for assessing the extent of angiogenesis and lesion vascularization, making it a key target for treatments aimed at reducing blood supply to endometriotic lesions. 

## 2. Materials and Methods

The study design is a prospective case-control cross-sectional study, which includes patients with endometriosis investigated and treated at the Obstetrics and Gynecology Clinic II of the Cluj-Napoca Emergency County Hospital, Romania, from January 2022 to December 2023. Additionally, a control group was recruited consisting of reproductive-age patients with non-endometriotic gynecological pathology, also treated at the Obstetrics and Gynecology Clinic II of the Cluj-Napoca Emergency County Hospital, Romania, during the same period. Informed consent was obtained from all subjects or their legal representatives by signing a detailed form regarding the investigated issue. Another consent form was signed by the subjects or their legal representatives for the use of personal data. The study was conducted following guidelines of the Helsinki Declaration and was approved by the Ethics Committee of the “Iuliu Hatieganu” University of Medicine and Pharmacy, Cluj-Napoca, Romania (AVZ251 dated 25 February 2022).

Inclusion criteria for the endometriosis group consisted of reproductive-age patients who met both clinical and paraclinical criteria for diagnosis of endometriosis, and after histopathological examination following surgical intervention, diagnosis of endometriosis was confirmed. As for the inclusion criteria for the control group, reproductive-age patients with non-endometriotic gynecological pathology who underwent surgical intervention were included. Exclusion criteria included:Patients who did not freely express their consent for enrollment in the study.Patients who did not meet the necessary conditions for study enrollment or who did not attend scheduled follow-up visits as per the protocol.

After obtaining informed consent for each person included in the study, three forms were completed:A form containing general data, family and personal medical history, symptomatologyQuestionnaire regarding the physical activity practiced by the patient.BSGE questionnaire and FSFI questionnaire.

The FSFI is a 19-item questionnaire designed as a concise, multidimensional self-report tool to assess the key dimensions of sexual function in women. It is psychometrically reliable, easy to administer, and has proven effective in distinguishing between clinical and non-clinical populations. This questionnaire was specifically created and validated to evaluate female sexual function and quality of life in clinical trials or epidemiological studies, and its continued use in these areas warrants further investigation.

Preoperatively, from each patient, regardless of group, a blood sample will be collected for analysis of certain biomarkers in the serum of patients from the endometriosis group and control group, respectively. The samples will be centrifuged for 15 min at 1000× *g* at a temperature ranging from 2 to 8 °C. Obtained supernatant will be stored at a temperature of −80 °C until processing. Additionally, intraoperatively, a tissue sample (ovarian cyst wall or leiomyoma fragment) will be obtained for analysis of biomarkers at this level. The tissue sample will be placed in saline solution and frozen at −80 °C until processing.

### 2.1. Analysis of Patient Symptoms and Endometriosis Severity

Endometriosis severity is assessed according to rASRM classification. The rASRM classification is designed to classify endometriosis via direct visualization of the pelvic organs at laparoscopy or laparotomy. This classification is based on assigning scores to endometriotic lesions found at the peritoneal and ovarian levels based on the size of the lesions. By analogy, points are also assigned for adhesions on the ovaries and uterine tubes. Additionally, points are given for partial or complete obliteration of the Douglas pouch. Finally, all assigned points are summed, and the resulting scores are classified into four severity classes.

Analysis of the patient’s symptomatology was assessed using a questionnaire proposed by the BSGE for pelvic pain. The first part of the questionnaire details the pain intensity on a scale from 0 to 10 and the type of pain: premenstrual, menstrual, non-cyclic pelvic, dyspareunia, abdominal pain related to the digestive tract during and outside menstruation, lower back pain, urinary bladder pain or pain during urination. The second part refers to impairment of intestinal function in the absence of gastrointestinal pathology, with impairment assessed as a sensation of incomplete bowel emptying, constipation, or traces of blood in stool. The next chapter investigates whether the patient underwent preoperative treatment (combined oral contraceptives, IUD, GnRH analogs, GnRH analogs + estrogen, progesterone, hormonal substitutes, or aromatase inhibitors). Fertility is also assessed within the questionnaire as

Unaffected,The patient trying to conceive for less than 18 months without success, orThe patient trying to conceive for more than 18 months without success.

The following questions refer to analgesic medication required by the patient (non-steroidal anti-inflammatory drugs [NSAIDs] or opioids) and history of surgical intervention for endometriosis. The last part of the questionnaire assesses the patient’s perceived health status on a scale from 0 to 100 and their daily activity. In our analysis, to conduct a more accurate statistical analysis, we considered the average pain described by the patient through the questions in the first part of the questionnaire.

Another element investigated in the group of patients affected by endometriosis was related to physical activity practiced by the patient. The variable is defined as the maximum value declared by the patient to 6 questions in the questionnaire regarding the frequency, duration, and intensity of sports activities and physical activities outside of sports. This variable is defined as SPORT_PHACT.

### 2.2. Analysis of Investigated Biomarkers

Both in the case of patients in the endometriosis group and in the case of patients in the control group, a blood sample was collected, centrifuged, and the resulting serum was frozen. Additionally, a tissue sample was collected and frozen. Subsequently, both samples were analyzed for expression of certain biomarkers (serum and tissue) believed to influence endometriosis development and progression. The studied markers were

IL 8—Elabscience Human IL-8 (Interleukin 8) Elisa kit, catalog no: E-EL-H6008IL 10—Elabscience Human IL-10 (Interleukin 10) Elisa kit, catalog no: E-EL-H6154VEGF—Elabscience Human VEGF-A (Vascular Endothelial Cell Growth Factor A) ELISA kit, catalog no: E-EL-H0111BDNF—Elabscience Human BDNF (Brain-Derived Neurotrophic Factor) ELISA kit, catalog No: E-EL-H0010.

### 2.3. Statistical Analysis

The study explored the differences in the values of certain variables (interleukins IL-8, IL-10, BDNF, and VEGF-A) between a sample of patients with endometriosis and a control group. These differences were tested using the Student’s *t*-test, designed for independent samples with unequal variances. To visualize these differences, box plots were employed, offering a clear graphical representation of the data distribution. The study utilized Ordinary Least Squares (OLS) linear regression to validate the effects of IL-8, IL-10, patient-reported pain, and physical activity on the severity of endometriosis. To minimize the risk of Omitted Variable Bias (OVB) in the regression analysis, several control factors were included: FSFI score, age, BMI, and the biomarkers BDNF and VEGF. The research aimed to ensure the reliability of the results by testing whether IL-8 and IL-10 maintain their sign and statistical significance in the regression, both with and without the presence of other biomarkers. This approach was adopted to mitigate the potential confounding effects of IL-8 and IL-10 taken from tissue samples. As a measure of robustness, the study also checked Pearson correlations between the severity of endometriosis and potential influencing factors. Additionally, scatter plot representations allowed for the observation of the linear shape of these correlations, further supporting the study’s findings.

## 3. Results

### 3.1. Patient Characteristics

In our study, a total of 46 Caucasian patients diagnosed with endometriosis who underwent surgical treatment at our clinic between January 2022 and December 2023 were included. In all 46 patients, the diagnosis of endometriosis was confirmed through histopathological examination. All patients in this study were initially informed about their participation. After signing the informed consent and GDPR agreement (The General Data Protection Regulation), patients responded to the questionnaire proposed by BSGE, and general data, family and personal medical history, symptomatology, and physical activity practiced by the patient were collected.

The mean age of patients included in this study was 28.25 years, with a standard deviation of 7.14. The severity of endometriosis, assessed by rASRM classification, had an average score of 26.50, corresponding to stage III severity: 5 cases were stage 2, 38 cases were stage 3, and 3 cases were stage 4. The average BMI of the patients was 25.31.

Regarding the control group, 44 patients met the criteria and were included. Among these, 36.36% (*n* = 16) had a histopathological diagnosis of uterine leiomyoma, 38.64% (*n* = 17) had a diagnosis of non-endometriotic ovarian cyst (serous cystadenoma, mucinous cystadenoma, luteal cyst), 20.45% (*n* = 9) had a diagnosis of mature ovarian teratoma, and 4.55% (*n* = 2) had a diagnosis of uterine septum. The mean age of the control group was 29.39 years.

### 3.2. Expression of Biomarkers at Serum and Tissue Levels

Analysis of biomarkers IL-8, IL-10, BDNF, and VEGF-A was conducted in both the endometriosis group and the control group with non-endometriotic gynecological pathology.

The expression of studied biomarkers in the endometriosis group is shown in Table 1, demonstrating their mean and standard deviation. Regarding IL-8 and IL-10, we observe a higher expression of these biomarkers at the tissue level compared to their serum level. On the other hand, BDNF and VEGF-A exhibit a higher expression in serum compared to their tissue level.

The same biomarkers were studied in the control group both at serum and tissue levels. Results are summarized in Table 2. In the case of BDNF and VEGF-A, the trend persists in the control group, with higher serum values compared to tissue expression. For IL-8 and IL-10, the trend also persists, but with a relatively smaller increase compared to the endometriosis group.

In the case of the control group, we analyzed mean values based on the pathology confirmed by histopathological examination. Table 3 summarizes the expression of biomarkers according to pathology confirmed by histopathological examination.

Analysis of biomarker values by diagnostic categories shows a certain pattern for each pathology, as follows:Uterine fibroid: exhibits elevated IL-8 values similar to endometriosis, with a similar trend for IL-10 but less elevated tissue values compared to endometriosis, higher BDNF values in both serum and tissue and lower serum VEGF-A values but higher tissue values.Non-endometriotic ovarian cyst: tissue expression of IL-8 and IL-10 had significantly lower values in cases of non-endometriotic ovarian cysts (43.15 vs. 114.36 pg/mL and 29.53 vs. 71.88 pg/mL, respectively). Regarding BDNF and VEGF-A, only the latter showed higher tissue values.Mature ovarian teratoma: shows reduced tissue values of IL-8 and IL-10 but higher values for BDNF and VEGF-A.

### 3.3. Results of BSGE Questionnaire

Regarding pain described by the patient, for statistical analysis, the average pain score was calculated based on patients’ responses to questions in the questionnaire. The mean pain score was 4.21 points on a numerical scale from 0 to 10, where 0 represents the absence of pain and 10 signifies the most severe pain imaginable, with a standard deviation of 1.24. Urinary tract involvement was identified in only 4 cases (8.7%), all of which reported suprapubic pain perceived at the level of the bladder, and the severity grade of endometriosis according to rASRM was 3. Intestinal function impairment was described in 17 cases (36.96%), with 5 patients reporting constipation and 12 patients reporting a sensation of incomplete bowel emptying. Intestinal involvement was predominantly encountered in patients with a more severe grade of endometriosis. Regarding medical treatment, 25 patients were using combined oral contraceptive (COC) or progestin medications for symptom control, and no patients were under treatment with Gonadotropin Releasing Hormones (GnRH) agonists, aromatase inhibitors, or other medications. Two patients were using levonorgestrel intrauterine devices (IUD) as a contraceptive method.

Patients’ history revealed that 34.78% of patients (*n* = 16) had a history of surgical interventions for endometriosis, and 6 patients presented non-gynecological comorbidities, including sinus tachycardia, bronchial asthma in 2 cases, chronic gastritis, hypertension, and type I diabetes. Regarding analgesic medication, 86.96% of patients reported using it, with 32.60% reporting chronic use outside of menstruation or premenstrual periods. Patients’ self-assessment of health on a scale from 0 to 100 showed a mean score of 77.07.

### 3.4. Correlation and Regression of Severity Determinants in Endometriosis

The correlation between values of studied biomarkers and the severity of endometriosis according to rASRM classification was analyzed. For a more detailed analysis, we used the score obtained within the classification rather than severity grade. The correlation between these variables and severity is shown in Table 4.

A statistically significant positive correlation was found between variables IL-8 tissue expression, IL-10 tissue expression, FSFI, and average pain score according to BSGE, with *p*-values below 0.05 (0.003, 0.005, and <0.001, respectively). A statistically significant correlation was also found in the case of hormonal treatment with combined oral contraceptives or progesterone and physical activity. In both cases, the severity of symptoms was lower with physical activity and hormonal medication use, with both having a *p*-value below 0.05 (0.003 and 0.042, respectively).

### 3.5. Analysis of Correlations between IL-8, IL-10 and the Severity of Endometriosis

Variables analyzed as determinants of endometriosis severity have a variable importance described according to Table 1. Furthermore, we analyzed the association between certain variables and endometriosis severity using different functions. Analyzing IL-8, we found that it is not statistically significant (*p* = 0.370) as a serum value, but its tissue expression is significantly higher in the endometriosis group compared to the control group (*p* = 0.003). In the control group, there is a patient with a very high IL-8 serum level. If we consider this value as an outlier and remove the patient from the group, the conclusion remains unchanged; the IL-8_S is not statistically significantly different in the endometriosis group compared to the control group (*p* = 0.713). These data are depicted in Figure 1a,b and Figure 2.

In the case of IL-10, its expression was significantly higher in tissue in the endometriosis group compared to the control group, while serum level did not show statistically significant values between the two groups. In the case of IL-10_S as well, there is a patient with an exceptionally high value, which can be considered an outlier. By removing this patient, the outcome remains unchanged; the difference between the two groups is not statistically significant (*p* = 0.473). The data are presented in Figure 3a,b and Figure 4.

The analysis of BDNF and VEGF-A between the two groups did not show statistically significant differences in the endometriosis group compared to the control group.

### 3.6. Analysis of Linear Regression of Patient-Reported Pain and Physical Activity on Severity of Endometriosis

The pelvic pain questionnaire proposed by BSGE was found to be directly correlated with the severity of endometriosis, as shown in Figure 5. Additionally, patients who used combined oral contraceptives or progestin medication had less extensive lesions at the time of surgical intervention.

Physical activity of patients has been shown to reduce the symptoms and also to have influence on the severity of the disease, as shown in Figure 6.

## 4. Discussion

The severity of endometriosis should be viewed beyond assessment through a numerical scale. While this scale allows for a concrete diagnosis and application of sequential treatment for the patient, the variability of the clinical picture makes it difficult to assess the patient’s prognosis. Therefore, a more detailed analysis of the factors contributing to the severity of endometriosis or those correlated with the severity of the disease is necessary. RASRM classification takes into account lesions present at the level of the peritoneum, ovaries, and Douglas pouch, as well as the presence of adhesions at the level of the ovaries or uterine tubes. The advantages of using rASRM are that it is the most widely used classification score for endometriosis, it is easy to use, and the four severity grades are easily understandable by patients. The disadvantages of using rASRM compared to ENZIAN are that it does not consider deep endometriosis, intestinal involvement or retroperitoneal lesions [28,29,30,31].

The existence of chronic inflammation in endometriosis is well-known and has been studied over the years. Studies by Taylor et al., Khorram et al., Orisaka et al., and many others have shown that the expression of cytokines such as IL-5, RANTES, or interferon-gamma (IFN-γ) is elevated and plays an important role in maintaining a state of chronic local inflammation [48,49,50,51,52,53].

IL-8 is a pro-inflammatory chemokine involved in recruiting and activating leukocytes at the site of inflammation and is produced by endothelial cells, monocytes, macrophages, and fibroblasts. However, the specificity of this interleukin is relatively low, as it is elevated in other diseases such as rheumatoid arthritis, psoriasis, inflammatory bowel disease, or cancer. This fact is also confirmed in our study by the fact that the serum IL-8 levels do not differ significantly between the two study groups. However, IL-8 can still be useful when analyzing tissue expression, as values are statistically significantly higher in the endometriosis group compared to the control group, demonstrating that tissue levels of IL-8 are directly proportional to the severity of endometriosis, a fact clearly visible in Figure 1 of our study. Pathophysiological implications of increased tissue expression of IL-8 include the presence of chronic inflammation leading to neovascularization, stimulation of pain receptors, decreased egg quality, and subsequent infertility.

These findings are corroborated by other studies in specialized literature that concluded that IL-8 plays a central role in the development of endometriosis through its pro-inflammatory function and neutrophil chemotaxis, being involved in all stages of lesion development (adhesion, invasion, implantation, and proliferation of ectopic endometrial cells), even protecting these cells against programmed cell death through apoptosis [54,55,56,57,58]. 

IL-10 is an anti-inflammatory cytokine produced by a variety of immune system cells, including T helper lymphocytes, monocytes, or macrophages. Its main functions include inflammation suppression and immunosuppression by suppressing the activity of immune cells, as well as promoting immune tolerance and immunomodulation [59]. IL-10 follows the same pattern both serologically and tissue-wise as IL-8, with non-significantly different serum levels between the two study groups but significantly different tissue levels between them, showing higher expression in endometriosis. From this, we conclude that stimulation of the local immune system (at the tissue level) is higher in cases of endometriotic lesions. Recalling the theories of endometriosis development, as mentioned in the introductory part, we specify that retrograde menstruation alone is not sufficient to explain the development of endometriotic lesions and that a certain immune cell deficit may be necessary. IL-10 is known for inhibiting the activation of T cells but also for reducing the expression of co-stimulatory molecules (CD-80 and CD-86) and indoleamine 2,3-dioxygenase. Interestingly, macrophages secrete IL-15, a chemotactic factor for uNK cells, and downregulate the cytotoxicity of uNK cells [59,60,61,62,63,64,65]. 

Tissue overexpression of IL-10 in association with other interleukins can explain immune tolerance, the deficit in phagocytosis of this cellular debris, and the adhesion and development of endometriotic lesions starting from these changes.

Surprisingly, the quantification of BDNF and VEGF-A in the two study groups did not show significant differences either serologically or in tissue. This could be explained either by the relatively small number of patients in the two study groups or by the variability of pathology in the control group.

It must be noted that the control group consists of patients with gynecological pathology and not disease-free patients. This fact limits the comparison of studied biomarkers between the two groups. Comparative studies of biomarkers’ expression in endometriosis and other gynecological pathologies could be the subject of future investigations.

Continuing analysis of biomarkers in the target group with endometriosis, we found high statistical significance for tissue values compared to their serum levels. Higher tissue values reflect a greater predictive accuracy and a direct correlation with the severity of endometriosis. Elevated tissue IL-8 reflects the local inflammatory process more accurately, providing an image of disease activity, combined with the presence of a deficit in the local immune system with immune tolerance to cellular debris reflected by the increased value of IL-10. It is worth mentioning again that serum levels of these biomarkers can be easily influenced by the presence of other systemic pathologies.

Patient-reported pain using the BSGE questionnaire has proven to be a faithful factor in predicting the severity of endometriosis. In addition to quantifying the pain experienced by the patient, the BSGE questionnaire investigates symptoms of urinary or gastrointestinal system involvement, as well as the impact on female fertility or daily routine activities. This can help us better understand the needs and concerns of patients and formulate more effective and personalized treatment strategies. Additionally, it allows monitoring the progression of the disease over time and evaluating the response to treatment.

The association between increased physical activity and a lower degree of severity of endometriosis confirms data from the literature. Regular physical activity and engaging in sports can reduce the severity of endometriosis because it helps lower estrogen levels, which play a key role in the development and progression of endometrial lesions. Exercise also enhances blood circulation, promoting the removal of excess hormones and toxins that may contribute to endometriosis symptoms. Furthermore, consistent physical activity is known to improve pain tolerance and reduce inflammation, offering relief from the chronic pain associated with endometriosis [66,67,68].

Our results are also supported by other authors in specialized literature. For example, Martire’s 2023 study emphasizes the role of differential diagnosis of types of dysmenorrhea reported by patients, highlighting in the diagnostic chapter the role of investigations regarding inflammatory and possibly autoimmune status for detecting endometriosis in young patients [69]. Of course, these investigations complement the classical diagnostic methods, with the same author emphasizing the role of transvaginal ultrasound performed by an expert in detecting early lesions of endometriosis [70].

Moderate or severe stages of endometriosis can also influence the course of pregnancy and the occurrence of pregnancy complications. The impact can involve multiple mechanisms, but one of the most current theories refers to the impairment of placentation development. This theory was also demonstrated by Salmeri in his 2022 study [71].

The present study, however, has several limitations and weaknesses. Verifying the results is best done by comparing the endometriosis group to a group of disease-free individuals without any other pathology. Given the lack of patients without any pathology to use as a control group, we decided that this would be the best way to validate our endometriosis group results. Among these, we mention:The relatively small number of patients in the two study groupsThe sample of patients exclusively of the Caucasian raceThe heterogeneity of the control group

## 5. Conclusions

In our study, we demonstrated that IL-8 and IL-10 can serve as a diagnostic and prognostic element of endometriosis, their value being directly proportional to the severity of endometriosis. These data confirm the presence of local chronic inflammation and the existence of an immune deficit that contributes to the development of endometriotic lesions. No relationship was identified between BDNF and VEGF-A serum or tissue levels and endometriosis severity. However, we cannot exclude their association with the pathophysiology of endometriosis development and various symptoms, as this analysis was not the focus of the current study. Future development of a panel of biomarkers capable of assessing the diagnosis and prognosis of endometriosis can be extremely useful.

The pelvic pain questionnaire proposed by BSGE showed a good correlation with the severity of endometriosis, and we believe that its introduction into clinical practice can enhance understanding of the disease, assessment of treatment response, and patient quality of life.

Once again, physical activity demonstrates its extremely important role not only in promoting a healthy lifestyle but also as an alternative or adjunctive treatment in alleviating endometriosis symptoms and disease regression.

To validate the results and to further understand endometriosis pathology, we propose conducting future studies focusing on the pathophysiological mechanisms, early diagnosis, and personalized treatment of patients.

## Figures and Tables

**Figure 1 diagnostics-14-01822-f001:**
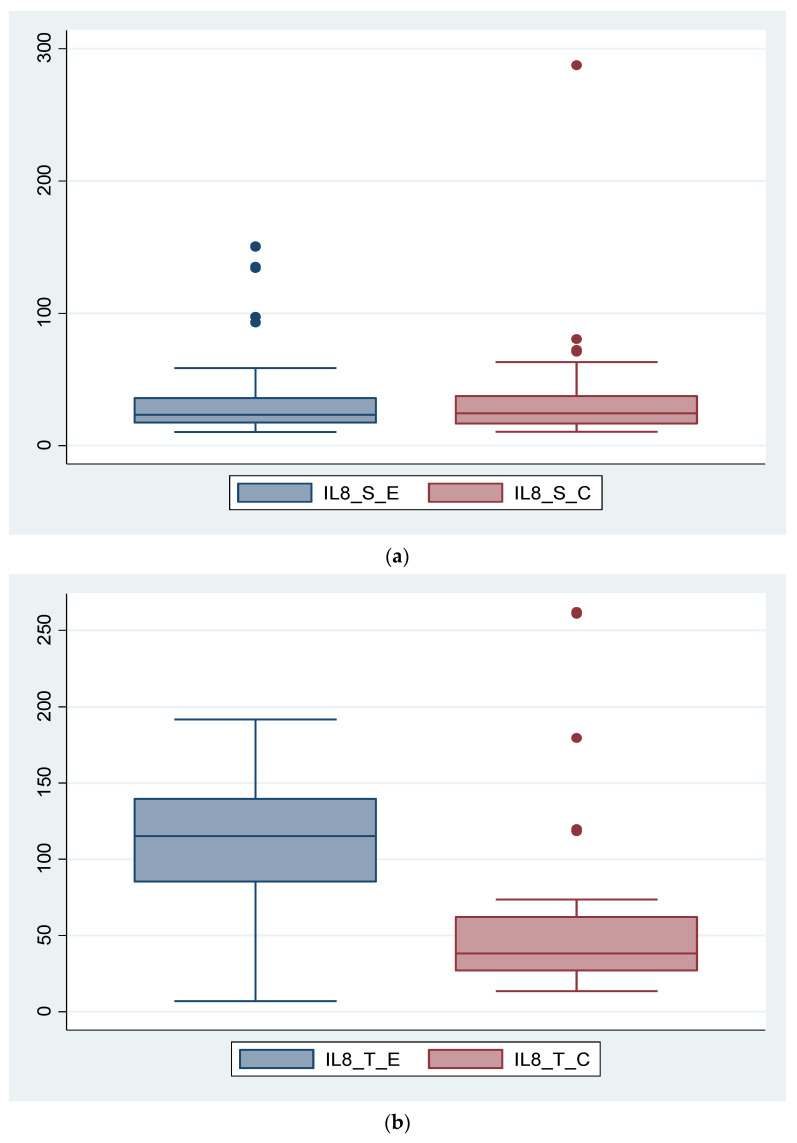
Comparative expression of IL-8 between endometriosis group and control group is illustrated as follows: (**a**) Serum expression of IL-8 in the two groups. (**b**) Tissue expression of IL-8 in the two groups.

**Figure 2 diagnostics-14-01822-f002:**
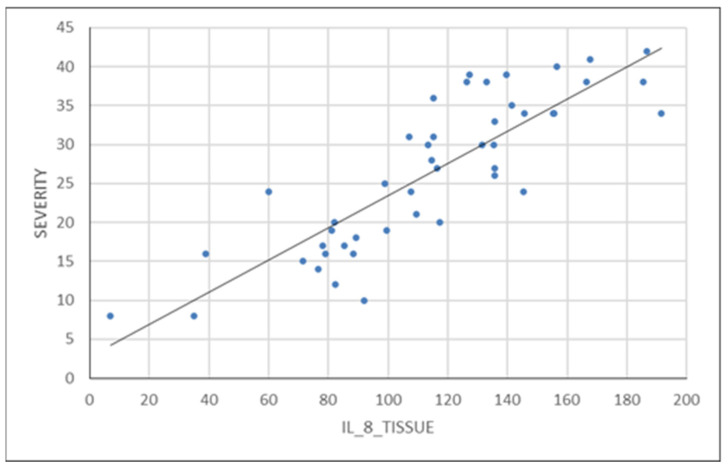
Correlation of tissue expression of IL-8 with severity of endometriosis.

**Figure 3 diagnostics-14-01822-f003:**
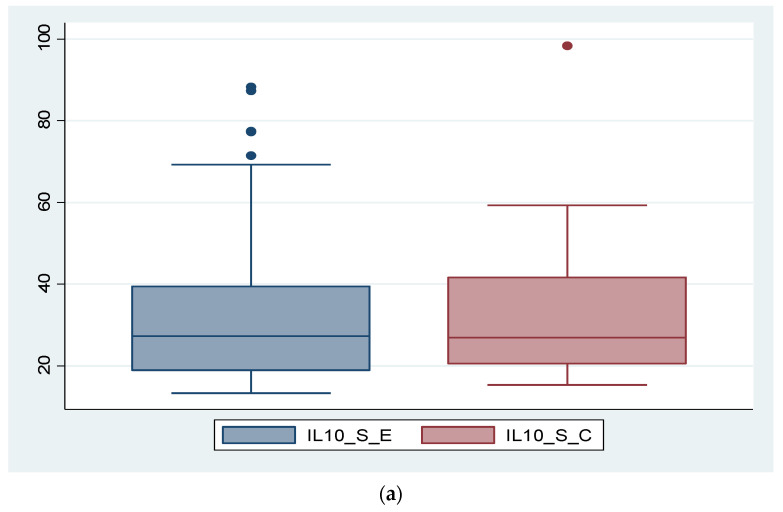
Expression of IL-10 compared between the endometriosis group and the control group: (**a**) serum expression of IL-10 in the two groups; (**b**) tissue expression of IL-10 in the two groups.

**Figure 4 diagnostics-14-01822-f004:**
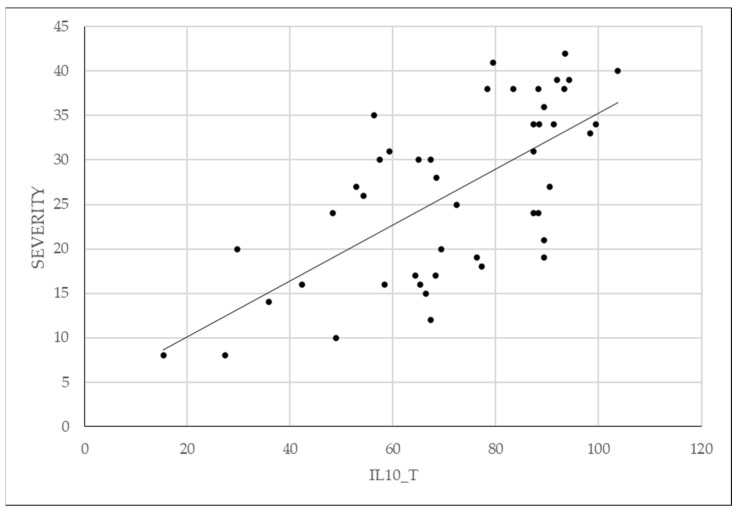
Correlation of tissue expression of IL-10 with severity of endometriosis.

**Figure 5 diagnostics-14-01822-f005:**
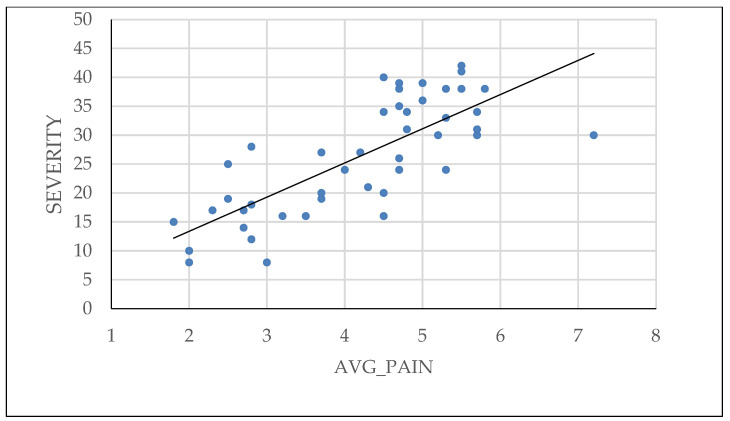
Linear regression of patient-reported pain and severity of endometriosis.

**Figure 6 diagnostics-14-01822-f006:**
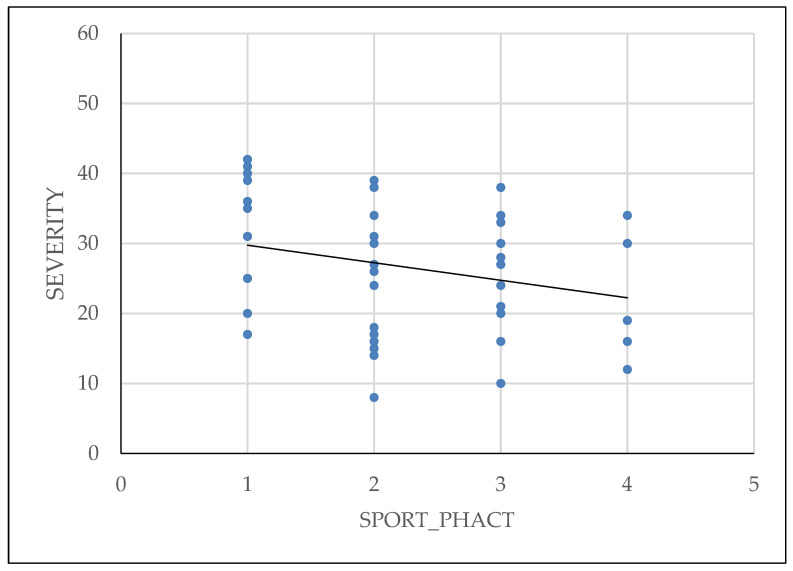
Linear regression of physical activity and severity of endometriosis.

**Table 1 diagnostics-14-01822-t001:** Expression of biomarkers at serum and tissue levels in endometriosis group.

	IL-8 S ^2^	IL 8-T ^3^	IL 10-S ^2^	IL-10 T ^3^	BDNF S ^2^	BDNF T ^3^	VEGF-A S ^2^	VEGF-A T ^3^
average	37.85	114.36	34.77	71.88	784.60	53.80	243.12	43.94
Std. dev. ^1^	36.93	39.5	20.75	20.69	505.77	25.32	185.53	19.86

Values expressed in pg/mL. ^1^ Standard deviation. ^2^ Serum levels. ^3^ Tisular levels

**Table 2 diagnostics-14-01822-t002:** Expression of biomarkers at serum and tissue levels in the control group.

	IL-8 S ^2^	IL-8 T ^3^	IL-10 S ^2^	IL-10 T ^3^	BDNF S ^2^	BDNF T ^3^	VEGF-A S ^2^	VEGF-A T ^3^
average	34.73	56.95	33.41	34.34	958.60	65.96	314.04	148.33
Std. dev. ^1^	42.02	54.28	18.84	15.65	639.74	48.85	378.49	120.44

Values expressed in pg/mL. ^1^ Standard deviation. ^2^ Serum levels. ^3^ Tisular levels.

**Table 3 diagnostics-14-01822-t003:** Expression of biomarkers at serum and tissue levels in the control group based on diagnosed pathology.

	IL 8 S ^1^	IL 8 T ^2^	IL 10 S ^1^	IL 10 T ^2^	BDNF S ^1^	BDNF T ^2^	VEGF-A S ^1^	VEGF-A T ^2^
uterine leiomyoma	36.42	100.99	37.97	48.89	994.31	99.05	167.81	240.58
non-endometriotic ovarian cyst	33.76	43.15	34.41	29.53	901.01	76.47	318.86	114.04
mature ovarian teratoma	25.29	32.91	37.57	23.78	1108.15	42.01	402.82	87.06
uterine septum	17.79	26.61	20.32	26.74	1215.63	70.78	215.19	97.67

Values expressed in pg/mL. ^1^ Serum levels. ^2^ Tissue levels.

**Table 4 diagnostics-14-01822-t004:** OLS regressions of several determinants on the severity of endometriosis (variable SEVERITY).

	Coef.	Std. Err.	*p*-Value	Coef.	Std. Err.	*p*-Value
IL-8_S				−0.027	0.036	0.464
IL-8_T	*** 0.065	0.021	0.003	*** 0.082	0.025	0.002
IL-10_S				0.035	0.065	0.591
IL-10_T	*** 0.094	0.032	0.005	** 0.087	0.038	0.029
BDNF_S				0.001	0.001	0.388
BDNF_T				−0.017	0.023	0.469
VEGF_S				−0.002	0.003	0.452
VEGF_T				0.042	0.030	0.175
FSFI	** −0.753	0.290	0.013	** 0.655	0.317	0.047
AVG_PAIN	*** 2.310	0.587	<0.001	*** 2.239	0.639	0.001
PROG_COC	*** −3.370	1.049	0.003	** −2.760	1.198	0.028
AGE	0.122	0.077	0.120	0.168	0.103	0.112
BMI	0.061	0.113	0.592	−0.074	0.140	0.602
SPORT_PHACT	** −1.243	0.589	0.042	** −1.682	0.718	0.026
	*n* = 46; Adj. R^2^ = 0.907 *n* = 46; Adj. R^2^ = 0.907					

***, ** significant at 1%, 5%. Source: own calculations in STATA 16.

## Data Availability

The data presented in this study are available on request from the corresponding author. The data are not publicly available due to internal rules.

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
