# Peer review of "Exploring the Influence of IL-8, IL-10, Patient-Reported Pain, and Physical Activity on Endometriosis Severity"

_diagnostics, 2024, doi:10.3390/diagnostics14161822_

Round 1

Reviewer 1 Report

Comments and Suggestions for Authors

Relevance

The relevance of this paper is notable due to the lack of reliable biomarkers in endometriosis diagnostics. The study provides insights into the roles of IL-8, IL-10, BDNF, VEGF-A serum and tissue levels, patient-related pain, and physical activity in a cohort of 46 patients diagnosed with endometriosis who underwent surgery, addressing a critical gap in current diagnostic approaches.

Areas for Improvement

Introduction:

The introduction section is somewhat lengthy. A more concise presentation would enhance readability and focus.

Emphasis on #ENZIAN:

Please explain why r-asrm was used? Why did you not use #ENZIAN? By using #ENZIAN the site of endometriosis and the correlating interleukines could provide intersting information. For example deep enodmtriosis vs peritoneal disease.

Figures:

Figures 1 and 3 lack units on the Y-axis, and the Y-axis in Figures 1, 3, and 5 needs improvement. The numbers and text are misaligned or overlapping, which hampers interpretation.

Discussion on IL-8 and IL-10 Usage:

The discussion section does not specify how IL-8 and IL-10 could be utilized in clinical practice. It would be beneficial to explore their use in monitoring endometriosis patients before and after interventions, during endocrine therapy, or in detecting recurrent disease. What are you ideas?

Author Response

Thank you for the evaluation and detailed feedback you provided on our article. We truly appreciate the time and expertise you invested in analyzing our work.

We have taken all of your comments seriously and are committed to addressing each aspect mentioned to improve the quality and relevance of our article. Point by point answer to reviewer comments are:

Introduction:

We attempted to shorten the introduction, but we believe that all aspects are particularly important for the reader to understand the role that these biomarker determinations can play in establishing the diagnosis and assessing prognosis of endometriosis.

 #ENZIAN

The rASRM classification was chosen over the ENZIAN classification because it is the one used in our medical institution.

Figures

We fixed fig. 1, 3 and 5

Discussion on IL-8 and IL-10 Usage

Serum or tissue determination of IL8 and IL10 can be used as an adjunct to the already known methods for diagnosing endometriosis. Furthermore, and perhaps more importantly, this determination performed at the time of surgery can help as a prognostic factor for the patient's risk of recurrence and quality of life. Scores that include these determinations as determining elements can be developed. All these can be useful for guiding the chronic treatment of patients with endometriosis. Lastly, biological therapies targeting these molecules can be developed to alleviate patients' symptoms and improve their quality of life.

Thank you once again for your valuable guidance. We are dedicated to improving this article and are grateful for your contribution to this process. We look forward to receiving your feedback in the future.

Best regards,

Andrei Malutan,  Ionel Nati

Reviewer 2 Report

Comments and Suggestions for Authors

There is a well-accepted disconnect between the ASRM stage of endometriosis and patient reporting of pain.  The intent of an endometriosis diagnostic on the basis of existing pain is to distinguish patients with pelvic pain and absence of disease from patients with pain and endometriosis. The alternative application of a diagnostic is to determine presence of endometriosis that is asymptomatic - no pain.  This report does not accomplish either.

The current authors apply an alternative pain scoring system (BSGE) that establishes a stronger association between BSGE pain scores and severity of disease. Main parameters of BSGE scoring should be highlighted. BSGE and FSFI scores are not specifically reported for the control group, so we cannot assess whether the pain scores among endometriosis patients are collectively greater or less heterogeneous than pain scores among control groups.  It is not clear that correlations of biomarkers IL-8 and IL-10 with any pain measure incorporate the control group (all data are reported as patients). BSGE scores (developed to predict surgical success for endometriosis) are not reported. Severity of pain by either measure and association with IL-8 or IL-10 have very modest correlation coefficients which suggest that this association does not comprehensively define the dominant factors that describe causes of pain. FSFI is undefined, so it is unclear what this means, and even if defined, the FSFI includes multiple assessments, and it is not clear what pain (menstrual, non-menstrual) is the significant driver of this correlation . The numbers of patients included are very small and there are no pain scores reported for the controls. The level of pain reported (2.3 on scale of 0-10 seems to integrate patients with mild pain). SPORT_PHACT is not defined so we do not know what is being measured.

Author Response

We sincerely thank you for the time and effort you dedicated evaluating our article. We appreciate the detailed and constructive feedback you provided. Point by point answer to reviewers comments as follows

About “There is a well-accepted disconnect between the ASRM stage of endometriosis and patient reporting of pain.  The intent of an endometriosis diagnostic on the basis of existing pain is to distinguish patients with pelvic pain and absence of disease from patients with pain and endometriosis. The alternative application of a diagnostic is to determine presence of endometriosis that is asymptomatic - no pain.  This report does not accomplish either.”

Serum or tissue determination of IL8 and IL10 can be used as an adjunct to the already known methods for diagnosing endometriosis. Furthermore, and perhaps more importantly, this determination performed at the time of surgery can help as a prognostic factor for the patient's risk of recurrence and quality of life. Scores that include these determinations as determining elements can be developed. All these can be useful for guiding the chronic treatment of patients with endometriosis. Lastly, biological therapies targeting these molecules can be developed to alleviate patients' symptoms and improve their quality of life.

About next paragraph:

In our study, the main focus is on the correlation between the proposed biomarkers and the severity of endometriosis. The control group consists of patients with various gynecological pathologies (uterine fibroids, non-endometriotic ovarian cyst, mature ovarian teratoma, uterine septum), and was reported in our article as a weakness of the present study. Our main focus was not to analyze the variability of biomarkers between the two groups of included patients, but rather to utilize the control group as a validation factor for the results of analysis of correlation between biomarkers and severity of endometriosis. In lack of of patients without any pathology to use as a control group we decided that thus  would be the best way to validate our endometriosis group results. For this reason, the BSGE and FSFI questionnaires were not applied to control group.

About BSGE, we added next paragraph in Introduction, page 2, lines 48-52: The main questions included in the BSGE questionnaire are related to: general question about your pain (Pre-menstrual pain, Menstrual pain, Non-cyclical pelvic pain, Pain during sexual intercourse, etc), Information about Bowel function, Medical Therapy, Fertility, painkillers,  previous surgery for endometriosis and about your health in general.

            About FSFI, we added next paragraph in Material and Metods, page 13, lines 404-309: The FSFI is a 19-item questionnaire designed as a concise, multidimensional self-report tool to assess the key dimensions of sexual function in women. It is psychometrically reliable, easy to administer, and has proven effective in distinguishing between clinical and non-clinical populations. This questionnaire was specifically created and validated to evaluate female sexual function and quality of life in clinical trials or epidemiological studies, and its continued use in these areas warrants further investigation.

            About SPORT_PHACT, we mentioned in page 14, lines 447- 450: Another element investigated in the group of patients affected by endometriosis was related to physical activity practiced by patient. The variable is defined as the maximum value declared by the patient to 6 questions in the questionnaire regarding the frequency, duration and intensity of sports activities and physical activities outside of sports. This variable is defined as SPORT_PHACT.

We truly appreciate the guidance and support you have provided us, and we are grateful for your contribution to improving the quality of our work. We look forward to receiving your feedback in the future, and we are open to any other suggestions you may have.

Sincerely,

Andrei Malutan, Ionel Nati
